# Extended 2,2′-Bipyrroles: New Monomers for Conjugated Polymers with Tailored Processability

**DOI:** 10.3390/polym11061068

**Published:** 2019-06-20

**Authors:** Robert Texidó, Gonzalo Anguera, Sergi Colominas, Salvador Borrós, David Sánchez-García

**Affiliations:** 1Grup d’Enginyeria de Materials (GEMAT), Institut Químic de Sarrià, Universitat Ramon Llull, Via Augusta, 390, 08017 Barcelona, Spain; roberttexidob@iqs.url.edu (R.T.); gonzaloanguerap@iqs.url.edu (G.A.); salvador.borros@iqs.url.edu (S.B.); 2Electrochemical Methods Laboratory—Analytical and Applied Chemistry Department at Institut Químic de Sarrià, Universitat Ramon Llull, Via Augusta, 390, 08017 Barcelona, Spain; sergi.colominas@iqs.url.edu; 3Centro de Investigación Biomédica en Red en Bioingeniería, Biomateriales y Nanomedicina (CIBER-BBN), 50018 Zaragoza, Spain

**Keywords:** pyrrole, 2,2′-bipyrrole, polypyrrole, conjugated polymer, electropolymerization

## Abstract

The synthesis of 2,2′-bipyrroles substituted at positions 5,5′ with pyrrolyl, *N*-methyl-pyrrolyl and thienyl groups and their application in the preparation of conducting polymers is reported herein. The preparation of these monomers consisted of two synthetic steps from a functionalized 2,2′-bipyrrole: Bromination of the corresponding 2,2′-bipyrrole followed by Suzuki or Stille couplings. These monomers display low oxidation potential compared to pyrrole because of the extended length of their conjugation pathway. The resulting monomers can be polymerized through oxidative/electropolymerization. Electrical conductivity and electrochromic properties of the electrodeposited polymeric films were evaluated using 4-point probe measurements and cyclic voltammetry to evaluate their applicability in electronics.

## 1. Introduction

Conducting polymers have demonstrated to be one of the most versatile sets of materials since their serendipitous discovery by Shirakawa and co-workers in 1977 [1]. The outstanding electrical and optical properties presented by these materials makes them excellent candidates for a large number of applications in the field of electronics. Conducting polymers have been extensively studied as part of energy storage systems [2,3,4], solar cells [5,6,7], electromagnetic interference shielding [8,9,10], or light-emitting diodes [11,12,13] among others. All these new systems within the framework of conventional electronics have allowed the appearance of new electronic devices such as smart windows, new display devices, or long-lasting batteries [14]. Recently, the versatility of the properties of the conductive polymers, such as their synthetically tunable electrochemical behavior, substrate adaptability, and biocompatibility has enabled devise innovative applications for biomedicine such as a new generation of biosensors [15,16], stretchable mechanical sensors [17,18], or stimuli-responsive biomaterials for tissue engineering [19,20]. Arguably, the most extensively studied conducting polymers are polyacetylenes (PA), polyaniline (PANI), polypyrrole (PPy), and polythiophenes (PTh). However, the latter polymers, PPy and PTh, have attracted the attention of many groups due to their outstanding properties as energy storage devices [21] or highly conductive materials [22].

From a synthetic standpoint, the preparation of such materials requires the careful design of the monomers. In general, many structural features of monomers contribute to the performance of the polymers. However, for conducting polymers obtained by electro-polymerization an important factor is the length of the conjugation pathway of the monomer, since it greatly determines the value of its oxidation potential. As it is well-known, the applied potential during electro-polymerization is a crucial parameter, since in some instances this process is carried out at potentials that cause polymer degradation, which leads to the formation of defects in the structure, lowering the conductivity and in some cases inhibiting the polymerization [23]. Another relevant structural feature of the monomers for producing good conducting materials is the presence of substituents at the β-positions of the corresponding aromatic units. The introduction of such substituents can impart solubility to the chains and modify the redox properties of the resulting materials.

In the case of polypyrroles, the use of substituted bipyrroles as monomers have provided polymers with enhanced conductivities, which have spurred their application in new fields. For instance, the use of bipyrroles such as 1,4-bis(pyrrol-2-yl)benzene as monomer has allowed polymerizations at low potentials preventing the degradation of the polymer backbone [23,24,25]. Similarly, the presence of alkyl and alkoxy substituents at the β-position of pyrrole introduce changes in the electrochromic [26] and redox properties of the resulting materials [26,27,28]. Unfortunately, the long and tedious syntheses of substituted pyrroles are a serious limitation for their application in the preparation of polymers. This is especially true for the synthesis of oligo(pyrrole-2,5-diyl) pyrroles (oligopyrroles).

Herein, we present a new family of chemically stable extended bipyrroles **1a**–**b** with α-free positions at the terminal pyrrole and thiophene rings. These compounds are prepared from easily available dibrominated 3,3′-disubstituted 2,2′-bipyrroles, which can be derivatized with pyrrole, *N*-pyrrole, thiophene groups at the 5,5′-positions by means of palladium cross-couplings (Figure 1). It is hypostatized that the extension of the conjugated pathway compared to pyrrole and bipyrrole will reduce the oxidation potential of the monomers. Furthermore, the substitution at the β-positions with ester and phenyl groups will favor the solubility of the resulting polymers in organic solvents. The effect of the reduction of the oxidation potential and the enhanced solubility of the monomers on the properties of the resulting polymers will be assessed in the present work.

## 2. Materials and Methods

### 2.1. Synthesis

All reagents and solvents of analytical grade were purchased directly from commercial sources and used without any further purification. Anhydrous solvents were drawn into syringes under the flow of dry N_2_ gas and directly transferred into the reaction flasks to avoid contamination. Column chromatography was carried out on silica gel (70–230 mesh) and analytical thin layer chromatography (TLC) was performed on plastic plates of silica gel GF-254 from Sigma Aldrich (San Louis, MO, US) with detection by UV. Standard techniques for synthesis were carried out under nitrogen atmosphere. All melting points were determined in a Büchi 530 capillary apparatus (Flawil, Switzerland) without any correction. Elemental analyses were performed on a Carlo-Erba CHNS-O/EA 1108 instrument (Val de Reuil, France). Compounds **1a** and **1b** were prepared according to reported methodologies [29,30]. Experiments were performed in an Initiator (Biotage, Uppsala, Sweden) microwave apparatus, operating at a frequency of 2.45 GHz with continuous irradiation power from 0 to 400 W. Reactions were carried out in 5 and 20 mL glass tubes (Biotage, Uppsala, Sweden), sealed with aluminium/Teflon crimp tops, which can be exposed to up to 250 °C and 20 bar internal pressure. The temperature was measured with an IR sensor on the outer surface of the process vial. After the irradiation period, the reaction vessel was cooled down rapidly to 50 °C by air jet cooling.

### 2.2. Electrochemistry

Electrochemical characterization and electropolymerization of 2,2′-bipyrrole monomers were performed using a three electrodes experimental setup: Pt foil was used as a working electrode, Pt mesh as a counter electrode, and a Pt wire in a glass tube with a porous membrane (filled with the supporting electrolyte solution) as a reference electrode. Monomers were dissolved (10 mM) in dry tetrahydrofuran (THF) with 0.1 M electrochemical grade tetra-butyl ammonium perchlorate (TBAP) as a supporting electrolyte. To avoid the formation of mixed materials during the polymerization and to improve the polymerization rate, a few drops of water were added to THF as proton scavenger. All electrochemical measurements were performed using a potentiostat/galvanostat Autolab PGSTAT302N (Herisau, Switzerland). The determination of the oxidation potential during electropolymerization of 2,2′-bipyrrole monomers was also measured against the ferrocene/ferrocenium couple (Appendix A).

### 2.3. Microscopy (FE-SEM)

The morphology of the polymeric films obtained through electrochemical polymerization was observed by Field Emission Scanning Electron Microscopy (FE-SEM) (MERLIN/Gemini II column from ZEISS, (Oberkochen, Germany). The polymeric films deposited on the platinum electrode were directly placed on microscopy supports for their characterization. No metallization or conductive coating was performed on the sample.

### 2.4. MALDI-TOF

The MALDI mass spectrometer used for the characterization of the bipyrrole derivate was a Microflex from Bruker (Billerica, Massachusetts, US). Samples were diluted in THF with a 1 µL of dithranol 0.1 M solution purchased from Sigma Aldrich (San Louis, Missouri, US).

### 2.5. Conductivity Characterization

Electrical conductivity measurements were performed trough four-point probe set up using a Keithley Sourcemeter 2600 (Beaverton, Oregon, US) with a spatial positioner to fix the electrode to the sample. The polymeric films were gently removed from the electrode using a conventional adhesive tape. For samples doping with iodine vapor, samples were also placed in an isolated flask with iodine. The exposition to the sublimated iodine completely oxidizes the polymeric chains enhancing electrical conductivity.

## 3. Results

### 3.1. Synthesis of the Monomers

The syntheses of monomers **1a**–**c** is predicated on the Suzuki and Stille couplings of dibrominated bipyrrole **2** and the corresponding boronic acid or stannilated derivative (Figure 2). In turn, the starting bromopyrrole is accessible from our previously reported methodology in two steps from ethyl cinnamate in an overall yield of 41% [30].

Once the brominated bipyrroles were in hand, preparation of substituted bipyrroles **1** was carried out by either Stille or Suzuki cross-coupling reactions. In particular, compounds **1a** and **1c** were synthesized following procedures previously reported by our group in 72% and 87% yield, respectively [29,30]. Extended bipyrrole **1b** was prepared by Stille coupling adapting the methodology used to prepare **1c**. The coupling afforded pure **1b** in 98% yield. This new compound was characterized by ^1^H, ^13^C-NMR, infrared spectroscopy and elemental analysis (EA) (cf. Appendix A).

### 3.2. Electropolymerization of 2,2′-Bipyrrole Monomers

The oxidation potentials of 2,2′-bipyrrole monomers were evaluated through cyclic voltammetry. The obtained values of the oxidation potential *E*_p,a_ against Fc^+^/Fc (Figure 3) were very similar when compared to the *E*_p,a_ values of pyrrole oligomers described in the literature [31]. As anticipated, a marked decrease in the oxidation potential was observed as the pyrrole oligomer increased in chain length (Table 1). For monomer **1a**, the first peak of oxidation is found at 0.17 V. This value is close to the one observed in the case of the unsubstituted quaterpyrrole (0.16 V). Interestingly, the methylation of terminal pyrrole rings (monomer **1b**) causes a decrease of the oxidation potential to 0.14 V. Monomer **1c** with external thiophene groups presents a *E*_p,a_ value of 0.25 V, which can be regarded as a low oxidation potential when compared with similar thiophene-pyrrole molecules [32].

As a preliminary experiment, the chemical oxidation of **1a**–**c** was attempted in order to assess the formation of polymeric materials. To do so, iron (III) chloride (0.1 M) was added as an oxidizing agent to a 10 mM solution of each bipyrrolic monomer in THF. As anticipated, in all cases, the addition of the oxidant produced changes in the solution color indicating that oxidative polymerization can be carried out. The solution was stirred for two days. Afterward, the solvent was removed and the crude material was washed with methanol. The residue was extracted with THF and the polymers were precipitated out by the addition of methanol. The synthesized polymers were fully characterized by ^1^H-NMR and GPC (Table 2).

Next, the electropolymerization of these monomers was examined in THF containing 0.1 M TBAP as a supporting electrolyte. Cyclic voltammetries were performed at 100 mV/s [33]. To obtain a polymeric film on the working electrode that could be easily characterized, a 10 × 10 mm platinum foil was used. Figure 3 shows the cyclic voltammograms of potentiodynamic electropolymerization of 2,2′-bipyrrole monomers: quaterpyrrole **1a**–**b** (Figure 3a,c) and bipyrrole **1c** (Figure 3b). In Figure 3a,b, the characteristic shape of typical conducting polymers during electrochemical synthesis can be seen, observing a higher current density on each scan due to the concentration of electroactive species in the media as expected for these kinds of monomers [34,35,36]. The presence of multiple redox couple peaks revealed that during the polymerization multi-electron transfer processes had been generated. The sudden increase of current density at 0.5 V for quaterpyrroles **1a** and **1b** and at 0.8 V dithienyl-substituted bipyrrole **1c** corresponds with the oxidation of terminal pyrrole or thiophene groups into radical cations and the consequent formation of the conjugated polymer [37,38].

The visual inspection of the working electrode during the polymerization of **1b** and **1c** revealed the formation of a homogeneous polymer film on the electrode surface. After 100 cycles, the differences between the measured current density in the two consecutive cycles were negligible indicating the termination of the polymerization. Unexpectedly, quaterpyrrole **1a** failed to form a film preventing the deposition on the platinum electrode. The close observation of the working electrode revealed that the polymeric film deposited during the oxidation step was dissolved in the THF media at the beginning of the reduction step (Figure 3d). This phenomenon indicates that the quaterpyrrole monomer was able to form a polymeric chain, but its solubility in the medium precluded its deposition on the working electrode. Interestingly, the presence of methyl groups at the terminal pyrroles produced a significant difference in the cyclic voltammograms during electropolymerization as can be observed in Figure 3c. The shape of the cyclic voltammograms showed a high resemblance with the one displayed by monomer **1b**; however, the characteristic increase of current density after each cycle was not observed in this case.

Matrix-assisted laser desorption ionization time-of-flight (MALDI-TOF) spectrometry was used to confirm the formation of oligomers during the electropolymerization [39,40,41]. Figure 4 shows MALDI-TOF spectra of the cyclic voltammetry media of each 2,2′-bipyrrole monomer revealing differences in the oligomer molecular weight distribution. Although complex parallel processes during electropolymerization may give rise to mixed materials [42], MALDI-TOF spectrums of the electropolymerization media revealed that oligomer formation was the prevailing process.

As expected, the spectrum for dithienyl-substituted bipyrrole **1c** (Figure 4a) exhibits its main peaks at *m/z* 1018.3 corresponding to the dimer and *m/z* 592.2 corresponding to the monomer. The absence of oligomers with n > 3 indicates that dithienyl-substituted bipyrrole was stable enough to deposit the higher polymeric chains on the working electrode. The higher intensity of the dimer peak may suggest that the dithienyl-substituted bipyrrole dimer should be especially soluble in the polymerization media or the monomer tendency to dimerize. The other peaks observed in the spectrum may be attributed to the combination of oligomers with other molecules presented in the cyclic voltammetry media as it is compiled in Appendix A.

Regarding the two monomers of quaterpyrrole, several differences were observed in the spectra during the cyclic voltammetry (cf Appendix A). Figure 4b shows the MALDI-TOF spectrum of quaterpyrrole **1b**, bearing nitrogen methylated groups in the terminal pyrroles, that presents great similarities with dithienyl-substituted bipyrrole (Figure 4a) where the main peaks are *m/z* 586.2 and 1170.1 corresponding with the monomer and dimer, respectively. A small peak at *m/z* 1755.8 corresponding to the n = 3 oligomer can be barely observed. Similarly, to **1c**, quaterpyrrole **1b** was successfully electrodeposited on the working electrode by electropolymerization. In contrast, the polymerization of **1a** shows a characteristic molecular weight distribution (Figure 4c) with clearly defined broad peaks for each oligomer. Thus, peaks at *m/z* 558.2, *m/z* 1116.4, *m/z* 1684.7, *m/z* 2227.9, and *m/z* 2786.1 precisely fit with the oligomer distribution from n = 1 to n = 5. The presence of oligomers of higher molecular weight than the other monomers is in accordance with the fact that quaterpyrrole **1a** does not form a film. Dissolution kinetic is much faster than the deposition kinetic preventing the polymeric chains to be deposited on the working electrode and obtaining a solution of 2,2′-bipyrrole-based oligomers.

It is interesting to note that the addition of two simple methyl groups in the monomer backbone produced such a dramatic effect on the solubility of the oligomer, providing the ability to generate stable 2,2′-bipyrrole films through electropolymerization or a solution of quaterpyrrole oligomers that eventually could be processed. That tunable properties through simple structure modifications reveal the promising role of 2,2′-bipyrrole monomers in the processing of conducting polymers.

### 3.3. Characterization of 2,2′-Bipyrrole Polymers Films

Morphologically, the prepared polymeric film presented a homogeneous and continuous structure. The images of the film obtained by the electrochemical polymerization of bipyrrole **1c** revealed 1 µm holes in some areas of its surface (Figure 5a); however, the rest of the observed surfaces displayed a continuous smooth surface (Figure 5c). In contrast, quaterpyrrole **1b** keeps its solid film homogeneity but with accentuated roughness (Figure 5b). A magnification of the image of the surface revealed that **1b** films may consist of clusters of <200 nm nanoparticles (Figure 5d).

Polymeric films were removed from the platinum working electrode using conventional adhesive tape. The conductivity of the samples was evaluated using a 4-point probe method. The results are shown in Table 3. The recorded thiophene substituted bipyrrole **1c** film conductivity similar to other thiophene derivates with alkyl-ester groups attached on the polymeric backbone that affects the electronic delocalization [43]. Quaterpyrrole **1b** films also present similar conductivity values when compared with conductive polypyrrole derivates [44,45] (between 0.1–100 S/cm depending on the derivative). These conductivity values reflect that, despite polypyrrole conductivity values not being enhanced through the monomer modification, the obtained polymers produce a continuous film. Continuity of the film morphology observed in the FESEM images for both samples, also validates that 2,2′-bipyrrole monomers were able to produce conductive polymeric films. The samples were also placed in an isolated flask with iodine in order to oxidize the polymeric chains to increase the conductivity as a well-known procedure [46]. The conductivity of the films, after the doping process, was increased by one order of magnitude for both cases, which is in agreement with the conventional behavior of conducting polymers.

The films corresponding to quaterpyrroles **1c** and **1b** were also evaluated by cyclic voltammetry in monomer-free THF 0.1M TBAP solution to test their electroactivity. Figure 6a,b show anodic and cathodic peaks of the electrochemical system with similar shapes of others found in the bibliography for polypyrrole-based films [47] or polythiophenes [48], respectively. Electrochromic behavior was observed during each new cycle in the cyclic voltammetry for both 2,2′-bipyrrole polymeric films. The film produced by compound **1b** showed a yellow appearance in a neutral state while during the oxidation step the color slowly changed to black (Figure 6c). Likewise, bipyrrole **1c** films showed a pale pink appearance in neutral state changing to transparent appearance during the oxidation step (Figure 6d). This electrochromic behavior is reversible for both films, revealing that 2,2′-bipyrrole monomers are able to provide conducting polymers with electrochromic properties.

## 4. Conclusions

In this study, stable extended 2,2′-bipyrrole has been prepared following a straightforward synthetic pathway. The resulting compounds were evaluated as monomers for the preparation of conjugated polymers. These oligomers display low oxidation potentials compared to pyrrole. This feature is ascribed to the higher extension of their conjugation pathway. The monomers are able to be polymerized through oxidative/electropolymerization obtaining a polymeric film deposited on a working electrode in the case of **1b**,**c** or dissolved in a processable organic solution in the case of the quaterpyrrole **1a**. The structure of the 2,2′-bipyrroles designed with the idea to minimize the oxidation potential during electropolymerization also presented the ability to modify the solubility of the polymeric chains. The obtained polymeric films presented interesting electrical and electrochromic properties when compared with similar conjugated polymers used in electronics. Future work will be devoted to the study of how including different substituent groups produces differences in solubility and, therefore, in processability.

## Figures and Tables

**Figure 1 polymers-11-01068-f001:**
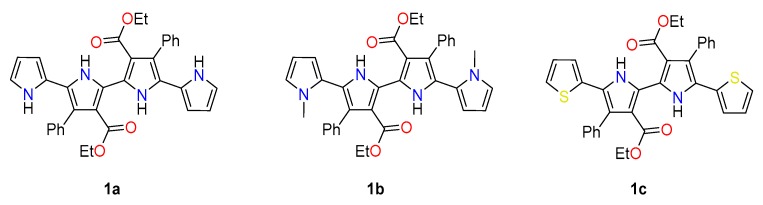
Structures of 2,2′-bipyrroles **1a**–**c**.

**Figure 2 polymers-11-01068-f002:**
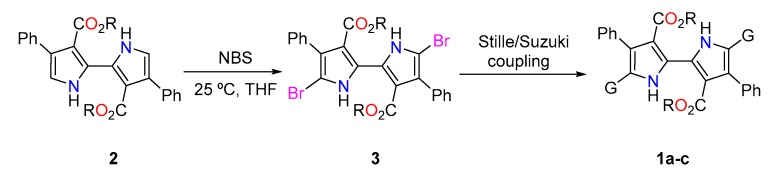
Synthesis of 2,2′-bipyrroles **1a–c** (G = 2-pyrrolyl, 2-(*N*-methylpyrrolyl) and 2-thienyl, R = Et).

**Figure 3 polymers-11-01068-f003:**
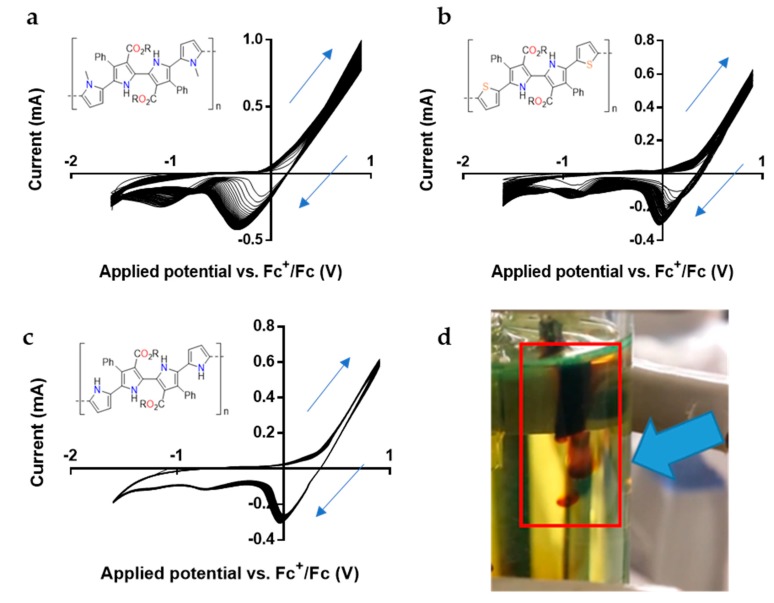
Cyclic voltammograms of 2,2′–bipyrrole monomers in THF (10 mM) + 0.1 M TBAP. The voltammograms correspond to **1b** (**a**), **1c** (**b**), and **1a** (**c**). Working electrode during quaterpyrrole electropolymerization (**d**) (R = Et).

**Figure 4 polymers-11-01068-f004:**
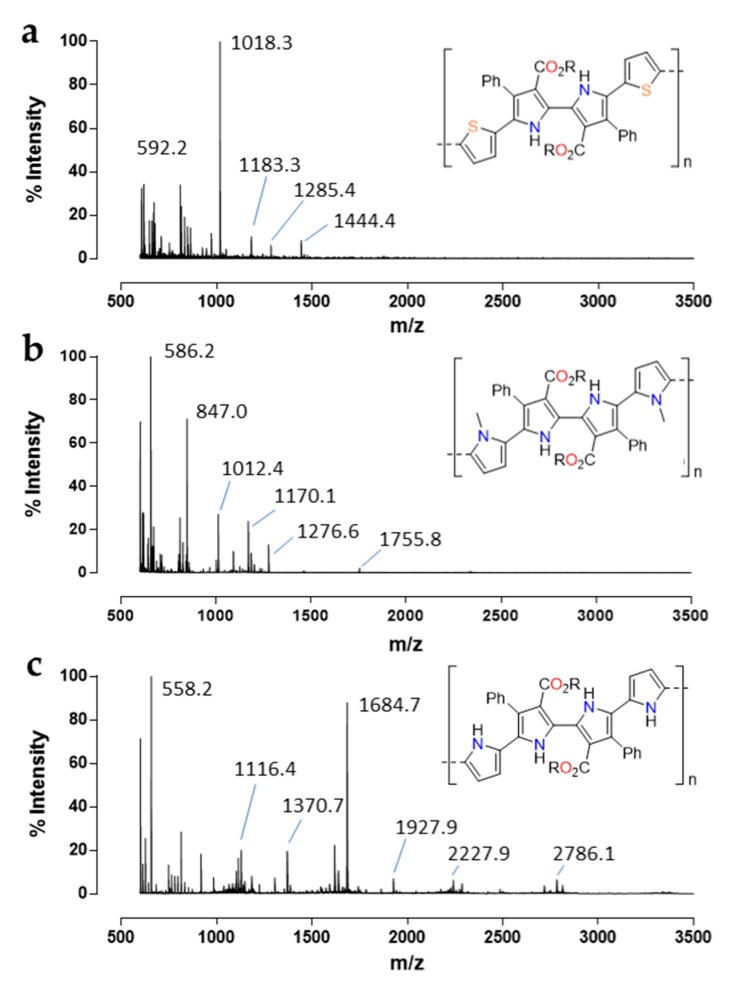
2,2′-Bipyrrole oligomers in the cyclic voltammetry media after 100 cycles shown by matrix-assisted laser desorption ionization time-of-flight (MALDI-TOF) spectrometry: **1c** (**a**) and quaterpyrroles **1a** and **1b** (**b**,**c**) (R = Et).

**Figure 5 polymers-11-01068-f005:**
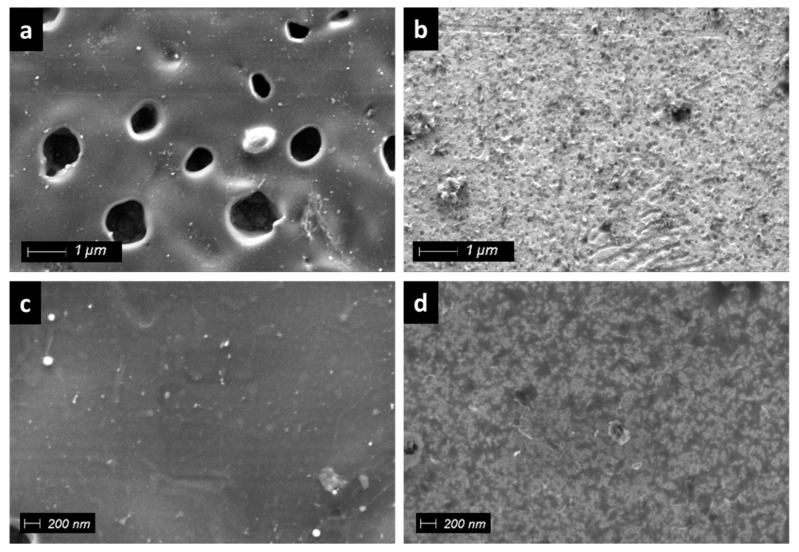
FE-SEM images of **1c** (**a**,**c**) and **1b** (**b**,**d**).

**Figure 6 polymers-11-01068-f006:**
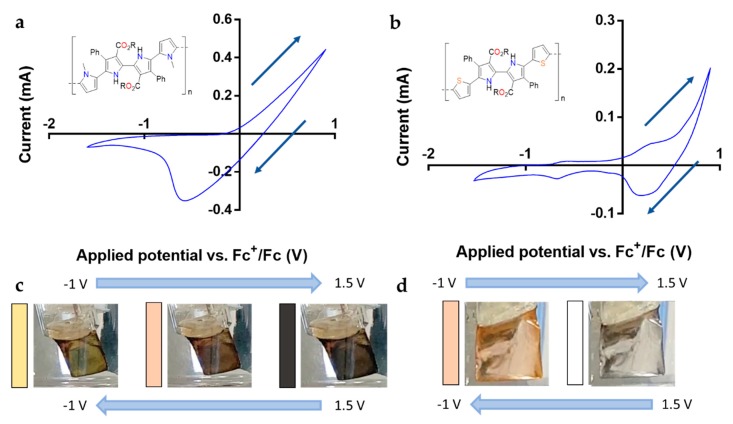
Monomer free voltammograms of **1b** (**a**) and **1c** (**b**). Electrochromic behavior of the films observed during the cyclic voltammetry of **1b** (**c**) and **1c** (**d**) (R = Et).

**Table 1 polymers-11-01068-t001:** Oxidation potential peaks (E)Pa against reference electrode and against Ferrocene for the 2,2′-bipyrrole monomers determined with the first cycles of electropolymerization.

Sample	*E*_p,a_ vs. Fc/V	Ref.
1*H*,-pyrrole	0.85	[29]
1*H*,1′*H*-2,2′-bipyrrole	0.6	[29]
1*H*,1′*H*,1″*H*-2,2′:5′,2″-terpyrrole	0.28	[29]
1*H*,1′*H*,1″*H*,1‴*H*-2,2′:5′,2″:5″,2‴-quaterpyrrole	0.16	[29]
**1a**	0.17	-
**1b**	0.14	-
**1c**	0.25	-

**Table 2 polymers-11-01068-t002:** Properties of weight, absorbance, and fluorescence of polymers prepared by chemical oxidation.

Monomer	*M* _w_ ^a^	*M* _n_ ^a^	PDI ^a^	Absorbance ^b^	Emission ^c^	Quantum Yield ^d^
**1a**	2300	1100	2.18	369	512	<1%
**1b**	2600	1100	2.46	408	499	0.04
**1c**	2300	1000	2.21	457	507	0.12

^a^ Estimated from GPC (eluent THF, polystyrene standards). ^b^ All spectra were recorded in THF at a concentration of 0.1 mg/mL. ^c^ Emission spectra were measured with excitation at the maximum absorption of each polymer. ^d^ Quantum yields were determined in THF using a solution of quinine in 0.05M H_2_SO_4_ (Φ_F_ = 0.546) as fluorescence standard.

**Table 3 polymers-11-01068-t003:** Conductivity of the films using the 4-point probe method before and after iodine doping.

Sample	S/cm
**1b**	5 × 10^−3^
**1c**	3 × 10^−4^
**1b** (iodine doped)	8 × 10^−2^
**1c** (iodine doped)	2 × 10^−3^

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
