# Peer review of "Extended 2,2′-Bipyrroles: New Monomers for Conjugated Polymers with Tailored Processability"

_polymers, 2019, doi:10.3390/polym11061068_

Reviewer 1 Report

The manuscript presents the synthesis of new monomers and new polymers. The polymers were generated by both, chemical and electrochemical monomeric oxidation.

- Could the authors present the electrochemical behavior of the polymers generated by chemical oxidation in solution after pressed to form an electrodic film or pill?

- The authors consider that the current flow only drives polymerization processes originating polymer films, soluble oligomers or a mixture of both processes. The literature has stated the presence of more complex parallel processes to give mixed materials, i.e. see the review Polym. Rev. 53, 311-351 (2013). Thus, the low reversibility of the polymer oxidation/reduction voltammetric processes and the presence of oligomer-derivatives in solution points to the presence of parallel chemical polymerization due to the strong pH variation around the electrode (two protons are liberated per monomeric unit incorporated to the polymer). Some drops of a proton scavenger (water) should improve the polymerization rate and the electro activity of the electrogenerated film. Other ways to improve the material properties can be open under consideration of a complex mechanism.

The authors have doped the electrogenerated polymer films with iodine vapor, does that means that the films were extracted from the elcetropolymerization cell after its electrochemical reduction? How the conductivity charge if the films are extracted from the solution after oxidation (doping) at 0.8 V for 60 seconds?

Arrows are required on the voltammetric responses indicating: the evolution of the potential along a voltammogram and the current evolution on the consecutive polymerization voltammetric responses.

Author Response

The manuscript presents the synthesis of new monomers and new polymers. The polymers were generated by both, chemical and electrochemical monomeric oxidation.

- Could the authors present the electrochemical behavior of the polymers generated by chemical oxidation in solution after pressed to form an electrodic film or pill?

We thank the reviewer for this suggestion. In this preliminary report, the chemical oxidation method has been only carried out to assess the feasibility of the polymerization. For this reason, we do not have evidence of the electrochemical behavior of the polymer obtained by the oxidative route. However, we plan to fully study these materials in the near future and thus perform such analysis.

- The authors consider that the current flow only drives polymerization processes originating polymer films, soluble oligomers or a mixture of both processes. The literature has stated the presence of more complex parallel processes to give mixed materials, i.e. see the review Polym. Rev. 53, 311-351 (2013). Thus, the low reversibility of the polymer oxidation/reduction voltammetric processes and the presence of oligomer-derivatives in solution points to the presence of parallel chemical polymerization due to the strong pH variation around the electrode (two protons are liberated per monomeric unit incorporated to the polymer). Some drops of a proton scavenger (water) should improve the polymerization rate and the electro activity of the electrogenerated film. Other ways to improve the material properties can be open under consideration of a complex mechanism.

We appreciate the advice and fully agree with the comment. In fact, we were aware of the formation of mixed materials during the electropolymerization. So, wet THF was used as solvent as suggested. In this way, we consider that the formation of mixed materials was minimized. Despite formation of mixed materials from parallel chemical polymerization cannot be excluded, MALDI-TOF spectrometry performed on electropolymerization media (Figure 4 and Table S2) seem to indicate that the main process during electropolymerization was the oligomer formation and/or the film deposition.

Considering your comment, we have extended the materials and methods section to reflect the presence of water in the solvent to improve the polymerization (lines 193-195):

“Despite complex parallel processes during electropolymerization may give rise to mixed materials, MALDI-TOF spectrums of the electropolymerization media revealed that oligomer formation was the prevailing process”

We have also extended the experimental part to reflect the possible formation of mixed materials. Polym. Rev. 53, 311-351 (2013) has been properly added to the article bibliography (reference 42).

The authors have doped the electrogenerated polymer films with iodine vapor, does that means that the films were extracted from the elcetropolymerization cell after its electrochemical reduction? How the conductivity charge if the films are extracted from the solution after oxidation (doping) at 0.8 V for 60 seconds?

For the doping process of the electrogenerated polymeric films, films were removed from the electrode after the electropolymerization process (100 cycles) ending the voltammetric cycle at the end of the oxidative stage. The film was removed using adhesive tape and was placed on an isolated flask, being exposed to iodine vapor as it is widely described. (For example, recently in Zhao, D et al. Highly conductive polythiophene films doped with chloroauric acid for dual-mode sensing of volatile organic amines and thiols. Sensors Actuators, B Chem. 2017, 243, 380–387)

In our experience, working with conducting polymers, we have observed that obtained polymeric films may not present a complete oxidation. Iodine vapor doping process assures the complete oxidation of polymeric chains enhancing the electrical conductivity as we have observed in:

Plasma Processes and Polymers. 9,485 – 490 (2012)

Plasma Processes and Polymers. 5,433 – 443 (2008)

Considering your comments, we have extended the materials and methods section to clarify the set-up of the doping process (lines 113-116):

“The polymeric films were gently removed from the electrode using a conventional adhesive tape. For samples doping with iodine vapor, samples were also placed in an isolated flask with iodine. The exposition to the sublimated iodine completely oxidizes the polymeric chains enhancing electrical conductivity”

Arrows are required on the voltammetric responses indicating: the evolution of the potential along a voltammogram and the current evolution on the consecutive polymerization voltammetric responses.

As you suggest arrows have been added to figure 3 and figure 6. Thank you for your advice.

Reviewer 2 Report

This paper reports some very interesting findings concerning the use of tetrameric aromatic units as monomers for conducting polymer synthesis.

The work is apparently well done, and the paper is well written. Citations are good (though perhaps some referencing of other related works to compare with the results here would be welcome). The paper could be published after the following comments have been taken into account.

The introduction could give more specific information about related conducting polymers, perhaps polypyrrole and polythiophene themselves (or derivatives), and reference to them would also be useful in the analysis of the conductivity and in the conclusion where it is stated that interesting properties are observed compared with other conducting polymers, but it doesn't say what these are.

Is there a hydrogen bond between the ester group and the pyrrole hydrogen atom? What influence might this have on the backbone of the polymer, and what consequence on the localisation of charge?  It is surprising that IR spectroscopy (or Raman for that matter) have not been done, as these techniques would provide valuable information on the neutral and doped materials.

In Figure 3 the direction of sweep of the voltammograms should be indicated.  The conformation of the molecules here is misleading.

For the electropolymerisation of 1c, it simply seems kinetically slower. What happens at longer sweep times? 

The authors mention that the N-methylpyrrole increases solubility, but presumably this is because of the conformational restrictions that group imposes. Has modelling been done? How does it compare with poly(N-methylpyrrole)?

Author Response

This paper reports some very interesting findings concerning the use of tetrameric aromatic units as monomers for conducting polymer synthesis.

The work is apparently well done, and the paper is well written. Citations are good (though perhaps some referencing of other related works to compare with the results here would be welcome). The paper could be published after the following comments have been taken into account.

We thank the appreciation of the referee for our work.

The introduction could give more specific information about related conducting polymers, perhaps polypyrrole and polythiophene themselves (or derivatives), and reference to them would also be useful in the analysis of the conductivity and in the conclusion where it is stated that interesting properties are observed compared with other conducting polymers, but it doesn't say what these are.

As suggested, a paragraph has been added to the introduction including some data about polypyrrole and polythiophene. Some comparative data in the electrical conductivity section has been added for discussion (lines 39-42).

“Arguably, the most extensively studied conducting polymers are polyacetylenes (PA), polyaniline (PANI), polypyrrole (PPy) and polythiophenes (PTh). However, the latter polymers, PPy and PTh, have attracted the attention of many groups due to its outstanding properties as energy storage device or highly conductive material”

Is there a hydrogen bond between the ester group and the pyrrole hydrogen atom? What influence might this have on the backbone of the polymer, and what consequence on the localisation of charge? It is surprising that IR spectroscopy (or Raman for that matter) have not been done, as these techniques would provide valuable information on the neutral and doped materials.

Yes, certainly. In fact, as can be observed in the crystal structure of the bipyrrole model shown below, a hydrogen bond is clearly formed between ester group and the pyrrole amine that provides flatness to the whole structure. Interestingly, the bithiophene moieties keep this flatness. A more, detailed study of the electronic structure will be published elsewhere. However, some insights of this matter ere published in J. Org. Chem. 2017, 82, 6904–6912.

In Figure 3 the direction of sweep of the voltammograms should be indicated. The conformation of the molecules here is misleading.

Changed as suggested.

For the electropolymerisation of 1c, it simply seems kinetically slower. What happens at longer sweep times?

We see your point with the kinetical behavior of monomers during the polymerization. Cyclic voltammetry of 1c was performed at different sweep times. It can be seen in upper figure that cathodic peaks current was affected by the sweep rate as expected by the Randles Sevcik equation. However, no significant change (new additional peaks) in shape was obtained when increasing it.

The authors mention that the N-methylpyrrole increases solubility, but presumably this is because of the conformational restrictions that group imposes. Has modelling been done? How does it compare with poly(N-methylpyrrole)?

We share the opinion of the referee. In fact, we were lucky to determine the crystal structure of the corresponding monomer. From this structure, it is clear that the presence of the methyl groups disturb the planarity of the molecule. This effect should account, in our opinion, for the enhancement of the solubility of both the monomer and the polymer.

Reviewer 3 Report

In this manuscript, Texido et al. report the design, synthesis and characterization of π-extended bipyrroroles and their polymerization to generate new conducting polymers. Through a rational design and synthesis in few synthetic steps, the authors generated new monomers which were further polymerized through electrochemistry. Interesting, as hypothesized by the authors, the extension of the π-conjugation in the precursors was shown to reduce oxidation, and the resulting oligomers were shown to be particularly promising in electrochromic devices.

The design and synthesis of new conjugated polymers is always an important trend in the materials chemistry literature. Despite being far from record-high values in terms of conductivities, and despite not achieving high molecular weight polymeric materials (in fact the authors are mostly obtaining trimers and tetramers), I think the results presented in this manuscript can be useful for the polymer community. The characterization of the materials has been well done by the authors and the writing is clear and straightforward. Overall, the authors presented a simple work that demonstrates the need of rational design of new conductive materials. Therefore, I think this manuscript will be suitable for publication in Polymers after addressing some minor points:

1- For clarity, I suggest the author to include the complete chemical structures of polymers 1a-c in the main manuscript. 

2- Similarly, I also suggest the author to include Table S1 in the main manuscript (MW). 

3- The reported molecular weights are a bit low in my opinion. Have the authors considered using their new monomers in classic Pd-catalyzed polymerization (Stille or Suzuki) to achieve higher molecular weights? Comparison between the two techniques and resulting materials would be particularly interesting to probe for the effect of molecular weight on the final properties. 

4- In terms of morphology, the author report the formation of “clusters” on the surface for 1b. Despite being purified, is there any proof that these nanoparticles are actually composed of polymer and not Fe impurities? I suggest the authors to consider doing EDX to probe for the nature of these nanoparticles. Analysis through AFM would also be interesting to look for phase and height morphologies in the solid-state. 

Author Response

In this manuscript, Texido et al. report the design, synthesis and characterization of π-extended bipyrroroles and their polymerization to generate new conducting polymers. Through a rational design and synthesis in few synthetic steps, the authors generated new monomers which were further polymerized through electrochemistry. Interesting, as hypothesized by the authors, the extension of the π-conjugation in the precursors was shown to reduce oxidation, and the resulting oligomers were shown to be particularly promising in electrochromic devices.

The design and synthesis of new conjugated polymers is always an important trend in the materials chemistry literature. Despite being far from record-high values in terms of conductivities, and despite not achieving high molecular weight polymeric materials (in fact the authors are mostly obtaining trimers and tetramers), I think the results presented in this manuscript can be useful for the polymer community. The characterization of the materials has been well done by the authors and the writing is clear and straightforward. Overall, the authors presented a simple work that demonstrates the need of rational design of new conductive materials. Therefore, I think this manuscript will be suitable for publication in Polymers after addressing some minor points:

We thank the appreciation of the referee for our work.

1- For clarity, I suggest the author to include the complete chemical structures of polymers 1a-c in the main manuscript.

As suggested, the chemical structure of the polymers has been included in manuscript (Figures 3, 4 and 6) and SI.

2- Similarly, I also suggest the author to include Table S1 in the main manuscript (MW).

Following your advice Table S1 has been included in the manuscript.

3- The reported molecular weights are a bit low in my opinion. Have the authors considered using their new monomers in classic Pd-catalyzed polymerization (Stille or Suzuki) to achieve higher molecular weights? Comparison between the two techniques and resulting materials would be particularly interesting to probe for the effect of molecular weight on the final properties.

Certainly. Currently we are working on a manuscript on this subject. The absorption spectra of some examples are shown below.

4- In terms of morphology, the author report the formation of “clusters” on the surface for 1b. Despite being purified, is there any proof that these nanoparticles are actually composed of polymer and not Fe impurities? I suggest the authors to consider doing EDX to probe for the nature of these nanoparticles. Analysis through AFM would also be interesting to look for phase and height morphologies in the solid-state

Thank you very much for your advice. The 1c Bipyrrole – bithiophene and 1b Bipyrrole – bipyrrole methylated polymeric film morphologies observed in Figure 5 were obtained through electropolymerization. At the start of this process the electrochemical media only contains monomer and the electrolyte (TBAP) in a THF solution. The electrochemical media do not contain any substance providing from Fe that may cause impurities modifying the substrate. It may happen in the samples obtained from oxidative polymerization, whose morphology have not been studied. The few numbers of species in the media during the polymerization allows to indicate that the origin of “clusters” in the surface of the films is polymeric.

We see your point; film morphology is important considering specific applications such as sensors or ionic membrane exchangers. However, this work is focused in the monomer design and synthesis to obtain new conjugated polymers. In this work, The microscopy images were used just to validate the formation of the film and provide a general idea of the morphology. We all agree with the importance of morphology characterization of these polymeric films and it will be discussed in a future work.
